# Advancing District Nursing Care Through a Learning Healthcare System: A Viewpoint on Key Requirements

**DOI:** 10.3390/healthcare12242576

**Published:** 2024-12-21

**Authors:** Jessica Veldhuizen, Marieke Schuurmans, Misja Mikkers, Nienke Bleijenberg

**Affiliations:** 1Research Centre for Healthy and Sustainable Living, Faculty of Health Care, University of Applied Sciences Utrecht, 3584 CS Utrecht, The Netherlands; 2Department of Education, University Medical Center Groningen, 9700 AB Groningen, The Netherlands; 3Department of General Practice and Nursing Science, Division Julius Center for Health Sciences and Primary Care, University Medical Center Utrecht, 3508 GA Utrecht, The Netherlands; 4Department of Health Technology & Services Research, University of Twente, 7500 AE Enschede, The Netherlands

**Keywords:** district nursing care, learning healthcare system, patient outcomes, integrated care, healthcare transformation, data-driven practice

## Abstract

The increasing complexity of healthcare needs driven by an ageing population places pressure on district nursing care. Many vulnerable older adults prefer to remain at home, requiring care coordinated with general practitioners and other professionals. This demand for integrated care is further challenged by a shortage of nursing professionals and the lack of standardised approaches to measure care quality. This article identifies the key requirements for implementing a learning healthcare system in district nursing care, using patient outcome data to foster continuous improvement and create a more adaptive, evidence-based, and patient-centred approach. This paper synthesises findings from multiple studies conducted as part of a PhD thesis, utilising a multi-method approach. These methods include examining patient outcomes in district nursing care and evaluating necessary cultural, organisational, and financial changes. Four key requirements were identified: (1) standardising patient outcome measures; (2) fostering a data-driven culture and strengthening professional autonomy; (3) enhancing organisational support and integrated care; and (4) adopting financing models that incentivise continuous learning and quality improvement. Implementing a learning healthcare system with patient outcome data in district nursing care requires a transformative shift. Standardising outcome measures, investing in information systems, and promoting continuous learning are crucial. Aligning financial incentives with patient outcomes, strengthening professional autonomy, and enhancing organisational support can make district nursing more responsive and capable of meeting complex needs. The described requirements are essential for advancing district nursing care through a more adaptive, evidence-based, and patient-centred approach.

## 1. Introduction

The quality, accessibility, and affordability of district nursing care are increasingly strained by the rapid ageing of the population, which brings a rise in complex, long-term healthcare needs. These patients often rely on a network of informal and formal care providers, including general practitioners, district nurses, pharmacies, paramedics, hospitals, and municipal services. The coordination of these diverse services within the patient’s home environment is crucial to ensuring effective and patient-centred care. Many older adults prefer to remain at home rather than in institutional care [1,2,3]. However, this growing demand for home-based care is further complicated by a shortage of qualified nursing professionals, exacerbated by burnout, an ageing workforce, and challenges in recruitment [4,5]. These pressures highlight the need for district nursing care as a speciality nursing practice to adapt to these evolving demands [6,7,8,9]. Additionally, the lack of robust evidence presents a significant barrier to understanding and enhancing the quality of district nursing care [10]. A potential strategy to guide and enable nurses to learn from their care practices and improve care quality is providing insights into patient outcomes. Evaluating patient outcomes is critical for understanding the effects of nursing care on individual patients [11]. Moreover, in the context of multiple challenges, measuring patient outcomes becomes essential to ensure the provision of high-quality, safe, effective, and patient-centred district nursing care [12]. While district nursing practice often focuses on organisational (structural) and care delivery (process) measures [13], there is a pressing need to emphasise patient outcomes, which reflect the impact of care on the health status of both individuals and the population.

In response to these challenges, there is growing recognition of the need for a more adaptive and integrated approach to healthcare delivery. The concept of a learning healthcare system (LHS) has emerged as a promising model. This model involves the continuous collection of patient data, the analysis of this information, and the application of these insights to refine healthcare practices, creating an ongoing cycle of learning and improvement [14,15,16,17]. The ultimate goal of an LHS is to create a healthcare system that is responsive to individual patient’s needs and capable of generating knowledge that can be applied to improve care at a population level. Experiences with the LHS demonstrate significant potential for improving care and patient outcomes. The literature highlights that an LHS can enhance data-driven decision-making, foster continuous learning, and improve care coordination across healthcare settings [15]. Although LHS implementation in primary care settings is still in its early stages [18], LHS environments are increasing globally, with demonstrated health benefits across multiple continents and a range of settings [19].

Additionally, specific insight into patient outcomes enables healthcare professionals to learn from their care delivery and improve their practices [11,20,21,22]. Incorporating patient outcomes into daily nursing practice is not a novel idea but a fundamental component of the nursing clinical reasoning process [23,24]. Additionally, the use of outcomes for learning and improvement has a long history, with the story of Florence Nightingale as an excellent example: Florence Nightingale, a pioneer in modern nursing, demonstrated the value of data-driven decision-making in healthcare. During the Crimean War, she systematically collected and analysed health outcomes to identify preventable causes of mortality, such as poor sanitation [25]. Her use of statistics and data visualisation helped her advocate for better hygiene practices, laying the foundation for outcome-based improvements in care.

Although patient outcomes hold great potential for enhancing district nursing care, their use in practice remains limited [26]. This is primarily due to a lack of robust evidence and the absence of clearly defined patient outcomes tailored to district nursing care [13,27], leaving nurses uncertain about how to leverage these outcomes to improve care. Addressing this gap is crucial for advancing the field.

In response, the central aim of this article is to identify the key requirements for implementing an LHS that can help overcome these barriers and advance district nursing care. An LHS offers a path forward by optimising care quality, fostering continuous learning, and addressing complex healthcare needs. This article draws on a series of empirical studies and a comprehensive literature review to explore how better patient outcome measurement, cultural shifts towards data-driven practice, enhanced collaboration, and restructured financing models can drive the transformation needed.

## 2. Materials and Methods

This paper is intended as a viewpoint article where we reflect on our past research studies and experiences [28]. This paper synthesises findings from multiple studies conducted as part of a PhD thesis, including data analysis, a mixed-methods study, a systematic review, a Delphi study, a national survey, and multi-method qualitative research [28]. The paper is composed of a short description of the conducted studies (under Materials and Methods; Section 2); the main findings of the conducted studies (under Results; Section 3); a reflection on the main findings and the implications for practice, policy, education, and research; and methodological considerations (under Discussion; Section 4). Various original research studies were conducted with different designs (Table 1). The original published papers describe the exact study designs, materials, and methods. All studies were conducted in the Netherlands.

### Participants

The definition, delivery, organisation, and funding of district nursing care vary globally [10,35,36]. This viewpoint article defines district nursing care as a holistic approach encompassing preventive, supportive, and rehabilitative care, including technical, psychosocial, and personal services provided by nurses to individuals and communities. This definition aligns with other European definitions [37,38,39] and reflects the scope of district nursing care in the Netherlands [6,40], where it supports independence, manages long-term conditions, and addresses acute illnesses. District nursing also plays a crucial role in coordinating care at home, linking clients’ needs with other healthcare professionals (e.g., general practitioners, physiotherapists) [6].

In 2021, approximately 139,500 professionals delivered district nursing care in the Netherlands, serving over half a million people, or 3.4% of the population [41,42]. The professional workforce in the Netherlands includes health aides, healthcare assistants, vocational nurses, district nurses, and specialised nurses, each with distinct roles and competencies [6]. Nursing aides assist patients with daily activities such as bathing, dressing, and mobility, focusing on basic personal care. Nursing assistants provide basic nursing care, including the administering of medication, wound care, and supporting patients with daily activities. Vocational nurses offer direct patient care, monitor the patient’s health status, administer treatments, and support rehabilitation. District nurses with a higher education degree lead and coordinate care in the community, providing advanced nursing care, managing complex cases, and supporting patients with chronic conditions or rehabilitation in their home environment.

## 3. Results

The findings from the various studies identified four key requirements for implementing an LHS that will advance district nursing care: (1) standardising the use of patient outcome measures; (2) fostering a data-driven practice culture and strengthening professional autonomy; (3) enhancing organisational support and promoting integrated care; and 4) adopting financing models that incentivise continuous learning and quality improvement.

### 3.1. Standardising the Use of Patient Outcome Measures

Currently, the use of patient outcomes in district nursing care as part of an LHS remains inconsistent [33]. One of the main barriers to the effective use of patient outcomes is the lack of standardised measurement tools specific to district nursing care [31]. In our nationwide survey study, it became evident that district nurses often rely on inconsistent, ad hoc methods for measuring outcomes, which limits their ability to aggregate data, compare results across different settings, or use outcomes as a basis for continuous learning and improvement [33]. For example, significant variation was found in how nurse-sensitive outcomes were measured, with some nurses using validated tools for outcomes like pain or delirium, while others used non-standardized methods or did not measure these outcomes at all [31,33]. Another significant challenge is the tension between the need for individualized care and the drive towards standardisation. Many district nurses express concerns that focusing too heavily on standardised outcomes could detract from the personalised nature of care [34]. However, other nurses state that using standardised outcomes is supportive for providing better care to patients [33,34]. Balancing the need for individualisation with the benefits of standardisation requires a flexible approach to outcome measurement. One potential solution is to adopt a core set of standardised outcomes relevant to all patients while allowing for the addition of individualised measures where appropriate. In this, it is important to clearly define and operationalize patient outcomes [34]. For example, standardised outcomes could include measures of quality of life, functional independence, or patient satisfaction [32], while individualised measures might focus on specific goals or concerns unique to each patient, such as the ability to manage a particular chronic condition.

### 3.2. Fostering a Data-Driven Practice Culture and Strengthening Professional Autonomy

The successful implementation of an LHS in district nursing care requires not only changes in how care is delivered but also a significant cultural shift within the nursing profession. Traditionally, many nurses have viewed the collection of patient outcomes and other data as an administrative burden rather than an integral part of their clinical practice [33,34]. This perception is compounded by high workloads, staff shortages, and the increasing complexity of patient care, all of which contribute to a reluctance to adopt data-driven approaches [33,34].

However, shifting the perception of data collection from an administrative task to a tool for improving patient care is essential for fostering a culture of continuous learning. This shift requires a concerted effort to demonstrate how data can be used to inform clinical decision-making, improve patient outcomes, and enhance professional satisfaction. For example, district nurses who viewed outcome measurement as a part of their daily practice reported that it allowed them to evaluate the effectiveness of their care interventions and make more informed decisions about their patient’s treatment plans [33,34]. These nurses saw outcome measurement not as an additional burden but also as a valuable tool for enhancing the quality of care they provided. Additionally, a data-driven approach can improve the recognition and appreciation of district nurses’ work. Before and during the COVID-19 pandemic, many district nurses felt undervalued, both within their organisations and by the wider public, despite their critical role in frontline care [30,34]. By systematically measuring and showcasing the positive impact of nursing interventions through patient outcomes, nurses can better demonstrate the value of their contributions to patient care and strengthen their professional standing [34]. This could boost morale and foster greater appreciation and trust in their expertise.

A crucial aspect of this cultural shift is strengthening the professional autonomy of district nurses [34]. Autonomy allows nurses to coordinate care based on data-driven insights, tailoring care to the specific needs of patients and collaborating effectively with other care providers. In this context, outcome data become tools that empower district nurses to make informed decisions and implement solutions that improve patient functioning and well-being [34]. Effective leadership is essential for embedding a data-driven culture within district nursing teams and organizations [34].

### 3.3. Enhancing Organisational Support and Promoting Integrated Care

To successfully implement an LHS in district nursing, healthcare organisations must proactively foster an environment that supports continuous learning and data-driven improvement [33,34]. Organisations must prioritise aligning their organisational goals with patient-centred outcomes, ensuring that the infrastructure, resources, and incentives are in place to support the long-term success of an LHS in district nursing [34].

It is crucial for organisations to provide nurses with the education and training they need to feel confident using data in their practice [33,34]. This includes technical training in using electronic health records (EHRs) and data collection tools and education on interpreting and applying data in clinical decision-making [34]. In many cases, nurses may lack the necessary skills to register, analyse, and interpret (outcome) data effectively, limiting their ability to use data for continuous learning and improvement [33,34]. It is important to keep in mind that nurses may need assistance from experts, such as data scientists or nurse scientists, to analyse outcomes [34]. However, nurses find it essential to take the lead in interpreting the results and selecting appropriate interventions in collaboration with the patient, ensuring they fit the patient’s situation [34]. Providing ongoing education and professional development opportunities focused on data registration, literacy, analytics, and interpretation can help address this gap and empower nurses to actively participate in an LHS.

Organisations must also invest in robust information systems that allow for the seamless collection, analysis, and sharing of patient outcomes across different healthcare settings [33,34]. This includes integrating EHRs and ensuring compatibility between systems used by different healthcare providers to facilitate collaboration. EHRs should provide insight into a complete picture of a patient’s care, including measured patient outcomes, thus facilitating better coordination and improving patient outcomes. Additionally, organisations must create a culture that values outcome measurement as a core care component, encouraging nurses to use data for reflection and practice improvement [34]. Providing ongoing professional development opportunities, particularly in data literacy and the use of outcome measures, is essential for equipping staff with the skills needed to thrive in an LHS.

Furthermore, the shift towards an LHS requires enhanced collaboration between district nurses and other healthcare providers, as district nursing care takes place within a broader care network [29,32]. Vulnerable older patients often depend on multiple care providers, including general practitioners, pharmacies, municipal support services, and hospitals, in addition to informal caregivers. Effective collaboration within this network is essential to ensure that care is coordinated, seamless, and tailored to the individual and sometimes complex needs of patients. However, organisational silos and differing priorities between care providers often hinder effective collaboration, which can result in fragmented care and missed opportunities for learning [30,34].

### 3.4. Adopting Financing Models That Incentivise Continuous Learning and Quality Improvement

A crucial barrier to implementing LHSs in district nursing care is the predominant focus on financial productivity, which is often prioritised by health insurers and district nursing organisations [34]. This focus on the number of care hours delivered per team can hinder nurses’ ability to use patient outcome data effectively and engage in continuous learning. Nurses expressed concerns that this excessive emphasis on productivity may result in penalties or fear of negative repercussions for low productivity or mistakes [34]. A shift towards financing models that encourage the integration of outcome measures and support continuous learning would be beneficial. Positive examples include good agreements between nursing organisations and health insurers, where focus on quality outcomes and supportive, trust-based relationships create an environment that enables the use of data in care delivery. However, the lack of sufficient investment, organisational support, and openness to using outcomes remains a significant barrier [34]. During the COVID-19 pandemic, it became also evident that there was insufficient (financial) investment given to district nursing care [30]. Financing models that foster collaboration and trust and provide resources to support nurses in utilising outcome data would help mitigate these challenges and enable district nursing care to evolve in a more evidence-based and adaptive direction.

## 4. Discussion

### 4.1. A Reflection on the Findings

Standardising patient outcome measures in district nursing care is essential for advancing the implementation of LHSs. Our findings revealed significant variation in how nurse-sensitive outcomes are measured. The inconsistent use of measurement tools limits the ability to aggregate data, compare results across settings, and leverage outcomes for continuous learning and improvement. At the same time, district nursing is inherently patient-centred, with care tailored to individual patients’ specific needs and preferences [43]. While this approach ensures personalised, high-quality care, it complicates implementing uniform outcome measures. Without a degree of standardisation, generating consistent data to inform broader quality improvement initiatives remains a significant challenge, hindering system-wide improvements and the development of evidence-based best practices [44]. Balancing individualisation with the benefits of standardisation requires a flexible approach. A potential solution is adopting a hybrid model that combines a core set of standardised outcomes—such as quality of life, functional independence, or patient satisfaction [32]—with individualised measures tailored to specific patient goals, such as managing a chronic condition. This ensures that care remains patient-centred while producing reliable, comparable data for system-wide learning. Additionally, recognising the multidimensional nature of health, which is often difficult to quantify, is crucial. Quantitative measurements should be complemented by qualitative insights, such as patient narratives, to ensure that data support and enhance professional judgment rather than replacing it [45,46,47].

Fostering a data-driven practice culture and strengthening professional autonomy is essential for implementing LHSs in district nursing care. Traditionally, high workloads and staff shortages have limited nurses’ engagement with data-driven approaches, despite professional guidelines emphasising the importance of documenting outcomes for quality care and clinical decision-making [33,34,48,49]. This requires a cultural shift, shifting the perception of (outcome) data collection from an administrative burden to a critical tool for improving care. This cultural shift also provides an opportunity to enhance the professional standing of district nurses. By systematically showcasing the impact of nursing interventions through patient outcomes, nurses can better demonstrate their value, increasing morale and trust in their expertise. Strengthening professional autonomy is another key element. Autonomy allows district nurses to use outcome data to tailor care to patient needs and collaborate effectively with others. Without sufficient autonomy, nurses may struggle to integrate data into practice, limiting the potential of an LHS. Empowering nurses to lead care coordination based on data-driven insights ensures their clinical judgment is trusted and valued [34]. Nurse leaders are instrumental in setting the tone, ensuring that data are collected and actively used to drive improvements [48,50].

Implementing an LHS in district nursing also requires strong organisational support and a commitment to integrated care. Divergent priorities among stakeholders—patients, nurses, organisations, payers, and governments—pose significant challenges to aligning efforts toward patient-centred outcomes [51,52,53]. Organisations must prioritise aligning goals with measurable patient outcomes and building infrastructure to support data-driven improvements, including resources, training, and incentives [33,34]. A key area for investment is workforce education. Many nurses lack the technical and analytical skills to effectively use outcome data [33,34,54,55]. Providing training in data registration, interpretation, and EHRs is essential [33,34,56]. Developing nurses’ data literacy is crucial for fostering their engagement in an LHS [56]. Organisations must foster a culture that values outcome measurement as a tool for continuous improvement while empowering and supporting nurse leaders to drive these initiatives [18,34]. Additionally, robust information systems are also vital for integrating care [56]. EHRs that enable seamless data sharing across healthcare settings will allow for better coordination and patient outcomes. However, current organisational silos and incompatible systems often hinder collaboration, leading to fragmented care [30,34,57]. Investing in integrated systems and improving communication across organisations will support a more cohesive approach to patient care. Integrated care is central to district nursing, particularly for vulnerable older adults who rely on multiple care providers [6,29]. Interprofessional education and training can enhance communication and teamwork across disciplines, fostering a collaborative environment that supports improved patient satisfaction and outcomes [52,58,59,60]. Although the global shift toward integrated care is promising, barriers such as communication gaps and insufficient financing must be addressed to achieve its full potential. With strong organisational support and a focus on integrated care, district nursing can harness the potential of LHSs to improve patient outcomes and system-wide learning.

To successfully implement LHSs in district nursing care, financing models must move beyond prioritising productivity and volume to actively support continuous learning, data-driven improvement, and patient-centred outcomes. Financing models play a pivotal role in shaping district nursing care, yet the current emphasis on productivity and volume often impedes the effective integration of patient outcomes into practice [46,61]. By prioritising the number of care hours delivered, existing fee-for-service models risk undervaluing the broader goals of continuous learning and quality improvement [46]. Transitioning to promising alternative payment models, such as outcome-based or bundled models, incentivises care providers to prioritise patient outcomes and quality improvements [46]. Such models encourage investments in integrated care, better coordination, and data-driven decision-making [46]. By linking financial rewards to long-term patient needs rather than short-term productivity, these approaches support district nurses in delivering personalised, evidence-based care while enhancing their professional autonomy [34]. Fostering trust-based agreements between nursing organisations and health insurers can create an environment that values quality outcomes over volume, enabling nurses to act on data-driven insights [34]. However, the shift to outcome-based financing presents challenges, such as defining standardised yet adaptable outcome measures and ensuring reliable data collection systems across care settings [62]. Policymakers are critical in addressing these barriers by creating supportive regulatory frameworks, funding pilot programs, and incentivising cross-organizational collaboration [57]. In doing so, financing models can evolve to align with the principles of a learning health system, driving improvements in patient care and supporting the sustainability of district nursing practices.

### 4.2. Recommendations for Practice, Policy, Education, and Research

A coordinated effort across practice, policy, education, and research is required to advance the implementation of LHSs in district nursing care.

In practice, district nurses must be empowered to participate actively in outcome measurement and data-driven decision-making. The coordination of care should be a shared responsibility between the district nurse and general practitioner, with both positioned as key coordinators. Providing them with the necessary training, tools, and data-driven insights is essential for managing complex patient needs effectively. By focusing on education and collaboration, supported by strong data systems, care teams can make informed decisions, improve care quality, and continuously learn from their practices, resulting in better integrated, patient-centred outcomes. Nurse leaders should foster a culture of continuous learning by regularly reviewing outcome data and encouraging staff to engage in quality improvement initiatives. Additionally, information systems must be optimised to facilitate data collection, analysis, and sharing while reducing administrative burdens. EHRs should be aligned with international standards like the “Nursing Process-Clinical Decision Support System” to integrate the nursing process and emphasise nurse-sensitive patient outcomes, ensuring consistency and reliability [63].

There is also an urgent need to highlight the importance of measuring patient outcomes in nursing education. Many practising nurses do not yet see the value or significance of outcome measurement for improving district nursing care. It is crucial to communicate that documenting patient outcomes is a core part of the nursing process, essential for delivering high-quality care rather than additional burdens. As one nurse stated in a focus group: *“If you have the right ones, then it’s not a burden but a delight... You take pride in them. That’s your value as a district nurse.”* [34]. To foster this understanding, a shift in motivation, attitudes, and behaviours is needed at all educational levels.

From a policy perspective, there is a need for reforms that support integrated care and outcome-based financing. Policymakers should work to align financial incentives with patient outcomes, moving away from productivity-based models that prioritise quantity over quality. Additionally, policies should promote collaboration between healthcare providers by creating regulatory frameworks that facilitate data sharing and joint decision-making. Moreover, both district nursing practice and policy initiatives should strongly advocate for uniform outcome measurements. A modest approach is recommended, starting with patient outcomes and measurement tools already familiar to nurses.

In education, nursing curricula should be revised to include training on data input, interpretation, and implementing changes based on the data, particularly for district nursing professionals. These skills are critical in executing key steps of an LHS. While district nurses should be equipped to handle these processes, a Chief Nursing Information Officer (CNIO) or data scientist could support the more complex data analysis tasks. The CNIO plays a crucial role in interpreting data on a broader organisational level and guiding the integration of data-driven insights into clinical practice [64]. By incorporating these skills into nursing education, future nurses will be better prepared to participate actively in an LHS. Continuing education programs should also be developed to provide practising nurses with opportunities to enhance their data literacy and improve their ability to leverage patient outcomes for continuous learning and practice improvement.

Finally, research into district nursing care is scarce [10,31]. Therefore, exploring the practical implementation of LHSs in district nursing care is needed. Pilot programs that test new outcome measurement tools, financing models, and collaborative care structures can provide valuable insights into what works and what does not. Additionally, further research is needed to understand the role of the Chief Nursing Information Officer (CNIO) in supporting data-driven practices and how this role can be expanded within district nursing care.

### 4.3. Methodological Considerations

This viewpoint paper utilised various research methods to provide valuable insights into how patient outcomes are used in district nursing care. However, several limitations emerged that must be addressed to strengthen future research and enhance its practical impact.

By employing systematic reviews, a Delphi study, national surveys, and multi-method qualitative approaches, the body of research incorporated multiple perspectives, which improved the overall validity of the findings discussed in this viewpoint article [65]. Despite this, the study faced challenges. The national survey had a low response rate, likely influenced by the COVID-19 pandemic and workforce shortages. At the same time, the systematic review’s focus on randomised controlled trials may have overlooked important insights from qualitative research. The lack of randomised trials in district nursing care emphasises the need for broader research methodologies, including non-experimental studies, to provide a more comprehensive understanding. This is underlined by the literature, as recent systematic reviews in district nursing care reveal a scarcity of randomized controlled trials [66,67,68], with most research comprising non-experimental quantitative, qualitative, or mixed-method studies, though these remain limited in both quantity and quality [69,70,71]. This highlights the need for a broader range of study designs to address knowledge gaps and strengthen evidence-based practices in district nursing care.

A key strength of the research was its close connection to real-world practice, involving district nurses, nurse assistants, and nursing students. Their involvement added practical relevance and increased the likelihood of successfully implementing the findings [72,73]. However, the studies were limited by the absence of direct participation from other crucial stakeholders, such as patients, policymakers, and organisational leaders. The lack of patient input, despite emphasising patient outcomes, may restrict the depth of understanding of patient preferences and needs. We did, however, incorporate patient insights by drawing on various reports regarding patient preferences, which were further verified by the Dutch Patients’ Federation [32]. Additionally, the limited involvement of broader healthcare stakeholders, particularly in areas requiring interprofessional collaboration, reduces the comprehensiveness of the findings. Future research should aim to include these key groups to enhance the applicability of the results and support more integrated care solutions.

## 5. Conclusions

Implementing an LHS, in which patient (outcome) data are used for learning and improving district nursing care, offers a promising pathway to improving care quality and addressing the increasingly complex needs of the ageing population. However, achieving this transformation requires the overcoming of significant barriers in patient outcome measurement, a cultural shift to embrace a data-driven practice, and the strengthening of the professional autonomy and coordinative role of the district nurses, organisational support, integrated care, and the financing of care. By empowering nurses to act on outcome data, make informed decisions, and coordinate care within the broader healthcare network, district nursing care can become more responsive to patient needs and more effective in improving long-term outcomes. Moreover, financial payment models should facilitate a transition to LHSs. Transitioning from fee-for-service models to outcome-based or bundled payment models will incentivize providers and nurses to prioritize care quality over quantity. Such models support better care coordination, enabling district nurses to implement patient-centred solutions that enhance patient well-being. Advancing district nursing care through an LHS requires the integration of outcome data into practice, the strengthening of professional autonomy, and the alignment of financial incentives with patient outcomes. By transitioning to a culture of continuous learning and collaboration, district nursing care can better meet the complex healthcare needs of the ageing population, ultimately leading to better patient care and more sustainable healthcare systems.

## Figures and Tables

**Table 1 healthcare-12-02576-t001:** Overview of research studies designs, materials, and methods.

Ref	Title	Objective	Design	Participants	Data Collection	Data Analysis
[29]	Predictors of district nursing care utilisation for community-living people in the Netherlands	To explore predictors of district nursing care utilisation for community-living (older) people in the Netherlands using claims data. To cope with growing demands in district nursing care, knowledge about the current utilisation of district nursing care is important.	Exploratory study using claims data	A total of 5500 pairs of community-living people using district nursing care and people not using district nursing care for two groups: all ages and age 75+ years (total N = 22,000).	Nationwide claims data from the Dutch risk adjustment system and national information system of health insurers.	The outcome was district nursing care utilisation, and the 114 potential predictors included predisposing factors, enabling factors, and need factors. The random forest algorithm was used to predict district nursing care utilisation. The performance of the models and the importance of predictors were calculated.
[30]	The Impact of COVID-19 from the Perspectives of Dutch District Nurses	To (1) explore, from the perspectives of Dutch district nurses, the impact of COVID-19 on patients who receive district nursing care, district nursing teams, and their organisations during the first outbreak in March 2020 as well as one year later and (2) to identify the needs of district nurses regarding future outbreaks.	Mixed-methods, exploratory study	District nurses (N = 36 for first phase, N = 18 for second phase).	The study followed a two-phase, sequential exploratory design, in which the results of the first qualitative method informed the second quantitative method (QUAL → quan). Phase 1: Semi-structured qualitative interviews. Phase 2: Online questionnaires.	Phase 1: Thematic analysis Phase 2: Descriptive statistics for closed questions; thematic analysis for open questions.
[31]	Evidence-based interventions and nurse-sensitive outcomes in district nursing care	To (1) provide an overview of interventions for community-living older people evaluated in district nursing care and evidence for the effects of these interventions, and (2) identify the nurse-sensitive outcomes that are used to evaluate these district nursing care interventions, how these outcomes are measured, and in which patient groups they are applied.	Systematic review	NA	Only experimental studies evaluating district nursing care interventions for community-living older people were included. Data sources were MEDLINE, CINAHL, PsycInfo, and EMBASE.	A data extraction form was developed to extract the study characteristics and evaluate interventions and nurse-sensitive outcomes. The methodological quality of the included studies was reviewed using the 13-item critical appraisal tool for randomized controlled trials by the Joanna Briggs Institute.
[32]	Nurse-sensitive outcomes in district nursing care	To determine nurse-sensitive outcomes in district nursing care for community-living older people. Nurse-sensitive outcomes are defined as patient outcomes that are relevant based on nurses’ scope and domain of practice and that are influenced by nursing inputs and interventions.	Delphi study	Experts with current or recent clinical experience as district nurses as well as expertise in research, teaching, practice, or policy in the area of district nursing (N = 15 round 1; N = 11 round 2).	The RAND/UCLA Appropriateness Method (RAM) was followed. Experts assessed potential nurse-sensitive outcomes for their sensitivity to nursing care by scoring the relevance of each outcome and the ability of the outcome to be influenced by nursing care on a nine-point Likert scale.	A group median of 7 to 9 indicated that the outcome was assessed as relevant and/or influenceable. To measure agreement among experts, the disagreement index was used, with a score of <1 indicating agreement.
[33]	Exploring nurse-sensitive patient outcomes in Dutch district nursing care	To explore (1) which nurse-sensitive patient outcomes are measured and how these outcomes are measured, (2) how district nurses use the outcomes to learn from and improve current practice, and (3) the barriers and facilitators to using outcomes in current district nursing practice.	Exploratory survey study	Vocationally trained and Bachelor-prepared registered district nurses (N = 132).	An exploratory cross-sectional survey study was conducted. The survey was distributed online among nurses working for various district nursing care organisations across the Netherlands.	Descriptive statistics for closed questions; thematic analysis for open questions.
[34]	Exploring the barriers, facilitators and needs to use patient outcomes in district nursing care	To provide in-depth insight into the barriers, facilitators, and needs of district nurses and nurse assistants on using patient outcomes in district nursing care.	Multi-method qualitative study	Vocationally trained and Bachelor-prepared registered district nurses in phase 1 (N = 132) *. Nurse assistants, vocationally trained and Bachelor-prepared registered district nurses in phase 2 (N = 26).	The approach consisted of sequentially utilising two qualitative data collection methods: initially, qualitative data were collected through open-ended questions in a survey, followed by in-depth online focus group interviews involving district nurses and nurse assistants in district nursing care in the Netherlands. The focus group interviews assumed a dominant role in the second phase (qual→QUAL).	Phase 1 and 2: thematic analysis.

* Notes: the participants in phase 1 of the multi-method study [34] are the same as those in the exploratory survey study [33].

## Data Availability

Not applicable.

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
