# Peer review of "Advancing District Nursing Care Through a Learning Healthcare System: A Viewpoint on Key Requirements"

_healthcare, 2024, doi:10.3390/healthcare12242576_

Round 1
Reviewer 1 Report
Comments and Suggestions for Authors
The research topic is important for promoting a practice in the field of nursing care.
The purpose of the article was to identify key requirements for implementing a learning healthcare system in district nursing care, using patient outcome data to foster continuous improvement and create a more adaptive, evidence-based, and patient-centred approach
Here are my comments after reading the article carefully:
1. Reading the method section in the abstract implies that this is a single study and not an analysis or focus of the findings of several studies. Only later does it say that they are part of a PhD thesis.
2. The introduction presents the issue comprehensively
3. Materials and Methods – Only at the beginning of this chapter is it stated that this is an article that presents a perspective on previous studies and experience as part of a PhD thesis. This is legitimate, but it should have been noted earlier. Beyond that, there is room for thought on how to write the rest of the article, for example presenting the findings. Furthermore, it is not clear from the presentation of the studies regarding the study/ies participants – for example, are the study participants in article 39 the same as the participants in article 40? Also, there is a great deal of variation between service recipients, professionals, and nurses... This point should be clarified.
4. Findings - the findings are presented including an explanation, a discussion of their interpretation, and even recommendations, for example line 152 or line 244. It is recommended to focus on the findings separately and present the interpretation in the discussion and recommendations section in an additional conclusions and recommendations section. This way, things will be clearer and more effective for the reader.
5. Section 3.4 It is not clear whether the section referring to financing models is based on research findings or is it part of a literature review on the subject?
6. The discussion section – deals only with recommendations and not with a discussion of the findings...
7. Acknowledgments - This section states (lines 435-436) that "this article serves as our synthesis of key findings and takeaways from the PhD thesis of the first author". This seems to answer at least one comme.
Author Response
The research topic is important for promoting a practice in the field of nursing care. The purpose of the article was to identify key requirements for implementing a learning healthcare system in district nursing care, using patient outcome data to foster continuous improvement and create a more adaptive, evidence-based, and patient-centred approach.
Reply: Thank you for recognising the importance of this research topic and for summarising the article's purpose so accurately. We appreciate your acknowledgement of the relevance of identifying key requirements for implementing a learning healthcare system in district nursing care.
Comment 1. Reading the method section in the abstract implies that this is a single study and not an analysis or focus of the findings of several studies. Only later does it say that they are part of a PhD thesis.
Reply 1: We appreciate your observation regarding the methods section in the abstract. Indeed, the findings presented in this paper are part of a broader PhD thesis. We acknowledge that this was not communicated as clearly as it could have been in the original text. To address this, we have revised the description of the methods in the abstract: “This paper synthesises findings from multiple studies conducted as part of a PhD thesis, utilising a multi-method approach.” We also changed the first sentences of Chapter 2, Materials and Methods, to make this more explicit. The revised sentence now reads: "This paper synthesises findings from multiple studies conducted as part of a PhD thesis, including a systematic review, a Delphi study, a national survey, and multi-method qualitative research."
Comment 2: The introduction presents the issue comprehensively
Reply 2: Thank you for your positive feedback on the introduction. We are glad it provides a clear and comprehensive overview of the research context.
Comment 3: Materials and Methods – Only at the beginning of this chapter is it stated that this is an article that presents a perspective on previous studies and experience as part of a PhD thesis. This is legitimate, but it should have been noted earlier. Beyond that, there is room for thought on how to write the rest of the article, for example presenting the findings. Furthermore, it is not clear from the presentation of the studies regarding the study/ies participants – for example, are the study participants in article 39 the same as the participants in article 40? Also, there is a great deal of variation between service recipients, professionals, and nurses... This point should be clarified.
Reply 3: Thank you for your insightful feedback. We have made several adjustments to address your comments:
- To enhance transparency, we now explicitly mention in the abstract that this article synthesizes findings from studies conducted as part of a PhD thesis. This clarification ensures that the reader understands the broader context of the research from the outset.
- We have revised the description of the structure of the results and discussion sections in the methods section. The results section now focuses exclusively on presenting the findings, while the discussion section provides the interpretation of those findings. We have also introduced a separate section outlining the recommendations for practice, policy, education, and research.
- In response to your comment about the participants, we have included a new paragraph in the methods section to clarify the overlap and variation in participants across the different studies. We now provide more detail regarding the diversity of participants (aides, assistants, nurses) involved in the research.
- Furthermore, to avoid confusion, it is now explicitly stated in the notes of the table whether the participants in different studies (e.g., article 39 vs. article 40) were the same or different. For instance, in the notes section: “The participants in Phase 1 of the multi-method study (40) are the same as those in the exploratory survey study (39).”
Comment 4: Findings - the findings are presented including an explanation, a discussion of their interpretation, and even recommendations, for example line 152 or line 244. It is recommended to focus on the findings separately and present the interpretation in the discussion and recommendations section in an additional conclusions and recommendations section. This way, things will be clearer and more effective for the reader.
Reply 4: Thank you for your suggestion. We have revised the structure as recommended. The findings are now presented separately, focusing solely on the results. The interpretation has been moved to the discussion, and the conclusions and recommendations are presented in a distinct section. We believe this enhances clarity and effectiveness for the reader.
Comment 5: Section 3.4 It is not clear whether the section referring to financing models is based on research findings or is it part of a literature review on the subject?
Reply 5: Thank you for your comment. The section referring to financing models was indeed based on both our own research findings and relevant literature. However, upon reflection, we realize that the emphasis on the literature review was too prominent. To address this, we have revised this paragraph to focus more on the findings from our own research.
Comment 6: The discussion section – deals only with recommendations and not with a discussion of the findings...
Reply 6: Thank you for your constructive feedback. We understand your concern regarding the focus of the discussion section. In response to your comment and in line with the feedback from other reviewers, we have revised the manuscript to ensure that the discussion section not only includes recommendations but also provides a more thorough analysis and interpretation of the findings. This revision now emphasizes a deeper exploration of the study's results, their implications, and how they relate to existing literature. We believe these adjustments improve the balance between presenting findings and offering recommendations, providing a more comprehensive discussion for the reader.
Comment 7: Acknowledgments - This section states (lines 435-436) that "this article serves as our synthesis of key findings and takeaways from the PhD thesis of the first author". This seems to answer at least one comme.
Reply 7: Thank you for your comment. We have ensured that the mention of the PhD thesis is clearly stated in key sections, such as the abstract, methods, and acknowledgements, to provide the necessary context. We trust this clarifies the reference without redundancy.
Reviewer 2 Report
Comments and Suggestions for Authors
Advancing district nursing care through a learning healthcare 2 system: a viewpoint on key requirements
Dear authors,
Thank you for the opportunity to review your interesting manuscript. First of all, I would like to congratulate all of you on this work, which effectively summarizes important and previous research.
I will give my feedback following the structure of the manuscript.
1.Title and abstract
The title is informative and the abstract provides a summary of the manuscript's major aspects. No further comments.
2.Introduction
The background chapter is clear, well-developed, well-structured, and thoroughly referenced. However, it lacks information on the actual outcomes of LSH model experiences across different contexts. In my opinion, the authors should address this point in the introduction section.
3.Methods
This section is clear and well-written. However, while it does not include specific details from the materials and methods section, it provides sufficient information to understand the structure of the current manuscript.
4. Results
I would like to congratulate the authors on this section, which is well-founded and well-structured. However, the format seems unusual to me, as it includes references to the studies presented and also discusses them in relation to other studies. At times, it reads more like a discussion than a results section. Therefore, the authors might consider moving some of this information to the discussion section and reducing the results section to focus solely on presenting the findings of these studies.
I also noticed some typographical errors, as some references appear in black font.
5.Discussion
I would like to congratulate the authors on this section. It is clear and effectively discusses the most relevant results of their study. However, as mentioned earlier, some information seems to be repeated from the results section. I encourage the authors to review both the results and discussion sections to avoid unnecessary repetition of information.
5.Conclusions
In my opinion, this section is very clear. Nothing to add.
Author Response
Thank you for the opportunity to review your interesting manuscript. First of all, I would like to congratulate all of you on this work, which effectively summarizes important and previous research.
Reply: Thank you for your kind words and for taking the time to review our manuscript. We appreciate your positive feedback and are glad that you found the work valuable in summarizing important research.
Comment 1.Title and abstract: The title is informative and the abstract provides a summary of the manuscript's major aspects. No further comments.
Reply 1: Thank you for your positive feedback on the title and abstract. We appreciate your recognition of the clarity in summarizing the key aspects of the manuscript.
Comment 2.Introduction: The background chapter is clear, well-developed, well-structured, and thoroughly referenced. However, it lacks information on the actual outcomes of LSH model experiences across different contexts. In my opinion, the authors should address this point in the introduction section.
Reply 2: Thank you for your valuable suggestion. We agree that incorporating experiences from different contexts would enhance the introduction. While we have not gone into detail, we have included references to studies that describe the experiences and increasing implementation of LHS models across multiple continents and settings, as noted by Foley et al. (2021), Nash (2021), and Enticott (2021). This approach allows us to maintain conciseness while still addressing the relevant experiences and demonstrated benefits in diverse healthcare environments.
Comment 3.Methods: This section is clear and well-written. However, while it does not include specific details from the materials and methods section, it provides sufficient information to understand the structure of the current manuscript.
Reply 3: Thank you for your positive feedback. We are glad that the methods section is clear and well-written. Based on feedback from another reviewer, we have made some revisions to improve clarity. Specifically, we have revised the description of the structure of the results and discussion sections in the methods. The results section now focuses exclusively on presenting the findings, while the interpretation of these findings is provided in the discussion section. Furthermore, we have expanded the description of study participants to clarify any potential overlap or variation, including more details on the diversity of participants across the studies.
Comment 4a. Results: I would like to congratulate the authors on this section, which is well-founded and well-structured. However, the format seems unusual to me, as it includes references to the studies presented and also discusses them in relation to other studies. At times, it reads more like a discussion than a results section. Therefore, the authors might consider moving some of this information to the discussion section and reducing the results section to focus solely on presenting the findings of these studies.
Reply 4a: Thank you for your positive feedback on this section. We appreciate your suggestion regarding the format of the results section. Based on your comment and the feedback from other reviewers, we have made revisions to the manuscript. Specifically, we have moved some of the interpretation and discussion of the studies to the discussion section, as you suggested. This adjustment allows the results section to focus solely on presenting the findings, providing a clearer distinction between the results and their interpretation. We believe these changes enhance the clarity and structure of the manuscript.
Comment 4b: I also noticed some typographical errors, as some references appear in black font.
Reply 4b: Thank you for pointing out the typographical errors regarding the references. We have carefully reviewed the manuscript and corrected these issues, ensuring that all references are consistently formatted. We appreciate your attention to detail and have taken steps to ensure the document is error-free.
Comment 5.Discussion: I would like to congratulate the authors on this section. It is clear and effectively discusses the most relevant results of their study. However, as mentioned earlier, some information seems to be repeated from the results section. I encourage the authors to review both the results and discussion sections to avoid unnecessary repetition of information.
Reply 5: Thank you for your positive feedback. We have already addressed the repetition between the results and discussion sections in comments 3 and 4. The results now focus solely on presenting the findings, while the discussion provides the interpretation.
Comment 6.Conclusions: In my opinion, this section is very clear. Nothing to add.
Reply 6: Thank you for your positive feedback. We are pleased to hear that the conclusion is clear and effective.
References:
- Foley T, Horwitz L, Zahran R. Realising the potential of learning health systems. Learn Healthc Proj. 2021;
- Nash DM, Bhimani Z, Rayner J, Zwarenstein M. Learning health systems in primary care: a systematic scoping review. BMC Fam Pract. 2021;22:1–13.
- Enticott J, Johnson A, Teede H. Learning health systems using data to drive healthcare improvement and impact: a systematic review. BMC Health Serv Res. 2021;21:1–16.
Round 2
Reviewer 1 Report
Comments and Suggestions for Authors
I appreciate the revisions made in accordance with the comments and recommendations given to the authors.
The article in its current form is much clearer.
Good luck!
Author Response
Thank you for your kind words and positive feedback on the revised manuscript. We are delighted to hear that the changes have improved the clarity of the article. Your thoughtful suggestions were invaluable in shaping the final version.
We greatly appreciate your support and encouragement, and we look forward to the next steps toward publication.